# Steering Control in Electric Power Steering Autonomous Vehicle Using Type-2 Fuzzy Logic Control and PI Control

**Bustanul Arifin** [1,2] , **Bhakti Yudho Suprapto** [1] , **Sri Arttini Dwi Prasetyowati** [2] **and Zainuddin Nawawi** [1,*]

1   Department of Electrical Engineering, Universitas Sriwijaya, Palembang 30128, Indonesia; bustanul@unissula.ac.id (B.A.); bhakti@ft.unsri.ac.id (B.Y.S.)
2   Department of Electrical Engineering, Universitas Islam Sultan Agung, Semarang 50112, Indonesia; arttini@unissula.ac.id
*   Correspondence: nawawi_z@unsri.ac.id; Tel.: +62-0771-310004

**Abstract:** The steering system in autonomous vehicles is an essential issue that must be addressed. Appropriate control will result in a smooth and risk-free steering system. Compared to other types of controls, type-2 fuzzy logic control has the advantage of dealing with uncertain inputs, which are common in autonomous vehicles. This paper proposes a novel method for the steering control of autonomous vehicles based on type-2 fuzzy logic control combined with PI control. The primary control, type-2 fuzzy logic control, has three inputs—distance, navigation, and speed. The fuzzy system's output is the steering angle value. This was used as input for the secondary control, PI control. This control is in charge of adjusting the motor's position as a manifestation of the steering angle. The study results applied to the EPS system of autonomous vehicles revealed that type-2 fuzzy logic control and PI control produced better and smoother control than type-1 fuzzy logic control and PI. The slightest disturbance in the type-1 fuzzy logic control showed a significant change in steering, while this did not occur in the type-2 fuzzy logic control. The results indicate that type-2 fuzzy logic control and PI control could be used for autonomous vehicles by maintaining the comfort and safety of the users.

**Keywords:** steering control; autonomous vehicle; type-2 fuzzy logic; PI control

## 1. Introduction

Humans are currently very dependent on modes of transportation to meet their needs, resulting in a rapidly growing number of vehicles on the road. According to WHO data, road accidents kill 1.3 million people each year [1]. Therefore, it is necessary to be careful when driving a car. One solution to overcome this issue is to use an autonomous vehicle [2]. With advanced features, this kind of vehicle can take someone to their destination with high navigation accuracy, optimal vehicle speed, and maintained comfort. Due to these benefits, many studies have been conducted dealing with autonomous vehicles. PID control has been used in vehicle steering control studies [3–6]. The stepper motor was used as an actuator in the Daewoo Matiz car. Closed-loop control generated smoother results than open-loop control [3]. The application of adaptive PID control to high-power servo motors was also used for autonomous vehicle steering [4]. Robust-PID control worked well on a midsized sedan, allowing it to follow the intended path [7]. PID control was also used to control the yaw rate and sideslip in lateral control [8].

A vehicle is made up of a series of subsystems. These subsystems include the engine propulsion subsystem, steering subsystem, braking subsystem, and others. It is a complex and nonlinear system. Controlling a system and analyzing the results require the use of mathematical modeling. Meanwhile, creating a vehicle model is not an easy task. The solution to this problem is to employ artificial intelligence. Unlike conventional controls, these modern controls do not require complicated modeling. Apart from being able to follow a predetermined path, an autonomous vehicle's stability factor while running

remains essential. A study on lateral control used a neural network to maintain yaw stability by minimizing the angle of deviation [9]. Vehicle lateral control uses a neural network based on a multilayer feed-forward method [10] and genetic algorithms [11]. Another solution for lateral control is fuzzy logic control (FLC). The authors of [12] used three inputs and two outputs for the lateral control of autonomous vehicles. The form of membership (MF) function used was a triangle and a 50-rule base, and the defuzzification used was the center of gravity method. With the various benefits of fuzzy logic, it is no longer necessary to model complex systems. The input in the study by [13] was the angle error and lateral error, and the output was the steering angle.

Fuzzy control outperforms conventional control [14], which is more efficient and robust. Parameter changes in the plant can always be followed properly by fuzzy control. The energy consumption of the system is lower. In addition, the conventional control requires a mathematical model to achieve satisfactory PID control, whereas FLC is no longer needed. According to the research by [15], FLC has better robustness than PID control for the vehicle control strategy. A comparison of conventional control and FLC was also carried out for the inverted pendulum robot [16]. The simulation results showed that the fuzzy logic controller outperformed the PID controller in terms of overshoot, settling time, and reaction to parameter changes. According to the research by [17], using FLC can improve the quality of the plant's output compared to using only model predictive control. If only using PID control, all of the control surfaces will affect each other when setting the parameters. Fuzzy logic control has the advantage of being able to overcome this conventional control's weakness. Using a rule base makes it possible to manipulate control surfaces individually. As a result, only the surrounding shape is affected, not all of the control surfaces [18]. Two advantages of fuzzy logic by other controls are the use of linguistic terms in the control process (both input and output) and the ability to efficiently handle uncertain inputs.

In comparison to type-1, type-2 FLC was selected for the machine reconfiguration process in order to avoid a detailed and complex modeling of the system parameter variations [19]. In type-1 FLC, the boundaries of the membership functions were fixed. In type-2 FLC, the boundaries of the membership functions were made uncertain. This has the potential to reduce the total number of rules in a type-2 rule base. The research by [20] on real swarm robots proved that type-2 FLC is more suitable and effective for control than type-1, since type-2 can handle and solve complex problems involving uncertainty to improve the accuracy. Type-2 FLC robots outperform type-1 FLC robots in terms of the ability to follow paths and avoid obstacles. When the uncertainty level was increased, autonomous robots that used type-2 FLC in the research by [21] were able to move faster and more precisely. Furthermore, type-2 FLC robots can move more smoothly than type-1 FLC robots. Type-1 logic control, on the other hand, requires precise knowledge for determining the input membership functions. Typically, the membership function is certain and accurate, so it must choose with certainty. It is difficult to determine the membership function. Type-2 fuzzy control uses a membership function between two values selected in a certain interval. This provides flexibility of the input values for control. In fuzzy control, this is called the footprint of uncertainty (FOU) [22]. Linguistic items are the main component of fuzzy logic. However, there are often differences between designers in translating them into antecedents and consequences, [23,24]. It is not easy to model the uncertainty caused by these differences, and it can reduce the system performance. Using type-2 fuzzy control can also simplify rule-based control [25–27].

The novelty of this study is a cascade control system that combines conventional with fuzzy logic control. Cascade control is used when the time constant between the control variable and the process variable is not short. This study employed cascade control in the form of nested loops. There was a DC motor that controlled the EPS steering system. The motor was controlled by a conventional proportional–integral control in the inner loop (secondary controller). The secondary control was directly linked to the plant, allowing it to track the rotation of the motor in order to reach the target position as quickly as

possible. The primary control used type-2 FLC. Type-2 fuzzy logic control is more robust to disturbances compared to type-1 fuzzy logic control. This is because type-2 FLC has a membership function with intervals, which allows it to be flexible. Autonomous vehicles have uncertain and non-linear variations in their parameters. Therefore, the determination of the required input parameters is included in a type-2 fuzzy membership function. The inputs used in this study were distance, navigation, and speed. The output of fuzzy logic control was fed into the PI control to run the steering system in the form of a DC motor. In this study, we expected to produce smoother and more stable control of an autonomous vehicle. The accuracy factor in reaching the predetermined trajectory and the passenger comfort factor is essential.

## 2. Lateral Vehicle Model

In the bicycle model, a front wheel at point A represents the car's two right and left front wheels. This also applies to the rear wheels. Figure 1 shows a vehicle modeled as a bicycle with two degrees of freedom [28]. The front wheel has the steering angles $\delta_f$, and the rear wheel has the steering angles $\delta_r$. This applies if both the front and rear wheels can be steered. Otherwise, the rear steering angle is zero. Point C is the center of gravity of the vehicle. The distance between the front and rear wheels is called the wheelbase (L), which is the sum of the distance A to cg ($\ell_r$) and the distance B to cg ($\ell_f$).

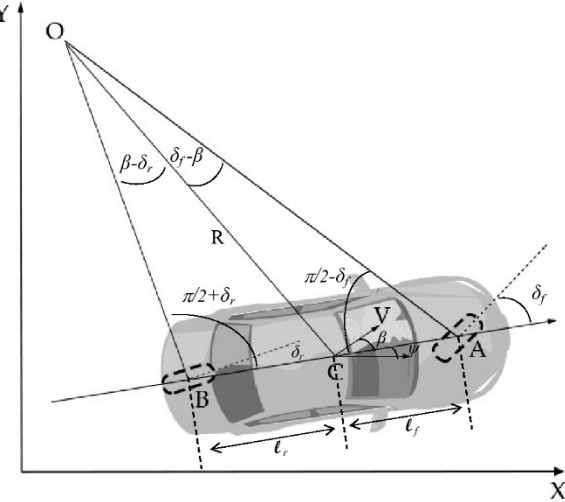

**Figure 1.** Lateral vehicle motion based on kinematics.

The vehicle motion is described in three coordinates—X, Y, and $\psi$. The inertial coordinates are shown on the X and Y, while the vehicle orientation is shown on $\psi$. The slip angle $\beta$ is the angle formed by the velocity of the vehicle at point C on the longitudinal axis.

The vehicle's instantaneous rolling center is point O. The perpendicular lines that intersect at point O are on both the front and rear wheels. The radius of the vehicle path (R) is given by the line connecting the cg to O. The vehicle's heading angle is called angle $\psi$.

According to the sine rule in triangle OCA, we get

$$\frac{\sin(\delta_f - \beta)}{\ell_f} = \frac{\sin(\frac{\pi}{2} - \delta_f)}{R} \tag{1}$$

$$\frac{\sin(\delta_f)\cos(\beta) - \sin(\beta)\cos(\delta_f)}{\ell_f} = \frac{\cos(\delta_f)}{R} \tag{2}$$

Equation (2) is multiplied by $\frac{\ell_f}{\cos(\delta_f)}$ on both sides become

$$\tan(\delta_f)\cos(\beta) - \sin(\beta) = \frac{\ell_f}{R} \tag{3}$$

According to the sine rule in triangle OCB, it gets

$$\frac{\sin\left(\beta - \delta_r\right)}{\ell_r} = \frac{\sin\left(\frac{\pi}{2} + \delta_r\right)}{R} \tag{4}$$

$$\frac{\cos\left(\delta_r\right)\sin\left(\beta\right) - \cos\left(\beta\right)\sin\left(\delta_r\right)}{\ell_r} = \frac{\cos\left(\delta_r\right)}{R} \tag{5}$$

Equation (5) is multiplied by $\frac{\ell_r}{\cos\left(\delta_r\right)}$ on both sides become

$$\sin\left(\beta\right) - \tan\left(\delta_r\right)\cos\left(\beta\right) = \frac{\ell_r}{R} \tag{6}$$

Equations (3) and (6) add up to

$$\left\{\tan\left(\delta_f\right) - \tan\left(\delta_r\right)\right\}\cos\left(\beta\right) = \frac{\ell_f + \ell_f}{R} \tag{7}$$

The rate of change of the vehicle orientation $(\dot{\psi})$ is equal to the angular velocity if the vehicle speed is assumed to be low. If the angular velocity is $\frac{V}{R}$, then

$$\dot{\psi} = \frac{V}{R} \tag{8}$$

Equation (7) is substituted into Equation (8) to get

$$\dot{\psi} = \frac{V\cos\left(\beta\right)}{\ell_f + \ell_r}\left(\tan\left(\delta_f\right) - \tan\left(\delta_r\right)\right) \tag{9}$$

The equation of motion can be written as whole as

$$\dot{X} = V\cos\left(\psi + \beta\right) \tag{10}$$

$$\dot{Y} = V\sin\left(\psi + \beta\right) \tag{11}$$

$$\dot{\psi} = \frac{V\cos\left(\beta\right)}{\ell_f + \ell_r}\left(\tan\left(\delta_f\right) - \tan\left(\delta_r\right)\right) \tag{12}$$

$$\beta = \tan^{-1}\left(\frac{\ell_f\,\tan\left(\delta_r\right) + \ell_r\,\tan\left(\delta_f\right)}{\ell_f + \ell_f}\right) \tag{13}$$

Figure 2 shows the new assumptions that must be applied when the vehicle is in a high-speed state and the tire slip-angle [29]. The development of dynamic models is required for the analysis of lateral vehicle motion. The vehicle lateral position $y$ and the vehicle yaw angle $\psi$ are representations of the system based on two degrees of freedom (DOF). Vehicle rotation is centered at point O for vehicle lateral position $y$. The $X$-axis is used to calculate the vehicle yaw angle $\psi$. The longitudinal velocity of the vehicle is defined as $V_x$.

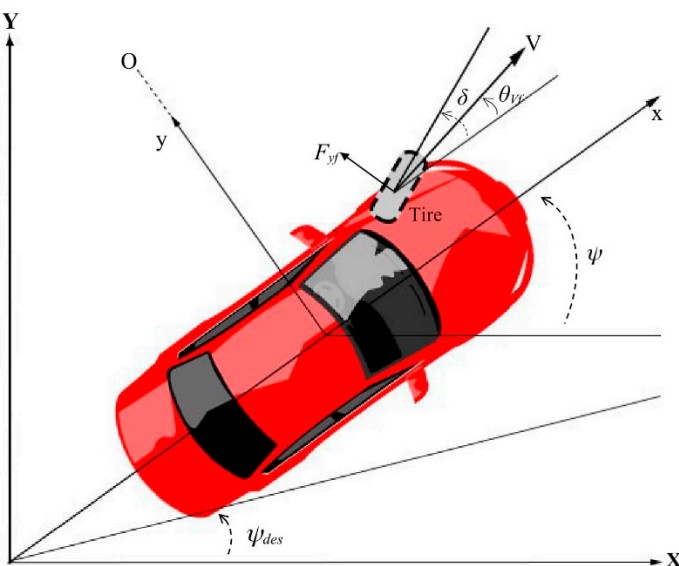

**Figure 2.** Lateral vehicle dynamics and tire slip angle.

The vehicle has an inertial acceleration at $a_y = \left(\frac{d^2 y}{dt^2}\right)$, while front ($F_{yf}$) and rear ($F_{yr}$) are the lateral tire forces. According to Newton's second law, it gets

$$ma_y = F_{yf} + F_{yr} \tag{14}$$

There is a motion along the $y$-axis that causes a $\ddot{y}$-acceleration and a centripetal acceleration $V_x \dot{\psi}$, so that

$$a_y = \ddot{y} + V_x \dot{\psi} \tag{15}$$

Then Equation (14) to be

$$m(\ddot{y} + V_x \dot{\psi}) = F_{yf} + F_{yr} \tag{16}$$

The yaw dynamic equation is obtained from the moment balance about the $z$-axis as

$$I_z \ddot{\psi} = \ell_f F_{yf} - \ell_r F_{yr} \tag{17}$$

where the distance between the cg and the front and rear tires is $\ell_f$ and $\ell_r$, respectively.

The angle of the vehicle between the velocity vector and the longitudinal axis is $\theta_V$, while the front wheel steering angle is $\delta$. Based on Figure 2, the slip angles of the front and rear wheels can be written as

$$\alpha_f = \delta - \theta_{Vf} \tag{18}$$

$$\alpha_r = -\theta_{Vr} \tag{19}$$

The cornering stiffness of each tire is $C_\alpha$, so the lateral tire force for the front and rear wheels can be written as

$$F_{yf} = 2C_{\alpha f}(\delta - \theta_{Vf}) \tag{20}$$

$$F_{yr} = 2C_{\alpha r}(-\theta_{Vr}) \tag{21}$$

To calculate $\theta_v$ on both wheels (front and rear), use the following equation:

$$\tan(\theta_{Vf}) = \frac{V_y + \ell_f \dot{\psi}}{V_x} \tag{22}$$

$$\tan(\theta_{Vf}) = \frac{V_y - \ell_f \dot{\psi}}{V_x} \tag{23}$$

Notation $V_y = \dot{y}$ and small angle approximation can be obtained:

$$\theta_{Vf} = \frac{\dot{y} + \ell_f \dot{\psi}}{V_x} \tag{24}$$

$$\theta_{Vr} = \frac{\dot{y} - \ell_r \dot{\psi}}{V_x} \tag{25}$$

The state space model can be obtained by substituting Equations (18), (19), (24) and (25) to Equations (16) and (17).

$$\frac{d}{dt} \begin{bmatrix} y \\ \dot{y} \\ \psi \\ \dot{\psi} \end{bmatrix} = \begin{bmatrix} 0 & 1 & 0 & 0 \\ 0 & -\left(\frac{2C_{\alpha f}+2C_{\alpha r}}{mV_x}\right) & 0 & -\left(V_x + \frac{2C_{\alpha f}\ell_f - 2C_{\alpha r}\ell_r}{mV_x}\right) \\ 0 & 0 & 0 & 1 \\ 0 & -\left(\frac{2\ell_f C_{\alpha f}-2\ell_r C_{\alpha r}}{I_z V_x}\right) & 0 & -\left(\frac{2\ell_f{}^2 C_{\alpha f}+2\ell_f{}^2 C_{\alpha r}}{I_z V_x}\right) \end{bmatrix} \begin{bmatrix} y \\ \dot{y} \\ \psi \\ \dot{\psi} \end{bmatrix} + \begin{bmatrix} 0 \\ \frac{2C_{\alpha f}}{m} \\ 0 \\ \frac{2\ell_f C_{\alpha f}}{I_z} \end{bmatrix} \delta \tag{26}$$

## 3. System Design

An autonomous vehicle was created using type-2 FLC, due to its advantages over its predecessor, type-1 FLC. It has three input parameters—distance, speed, and navigation. Fuzzy control produces an output in the form of an angle. Furthermore, this controller is called the primary controller. This value is used as an input by the motor controller, which is in charge of moving the vehicle's wheels to the right, straight, or left. This part is called the secondary controller. The encoder is used to provide feedback from the motor movement. The vehicle was equipped with sensors, such as proximity sensors, speed sensors, and navigation sensors. Figure 3 shows a block diagram of a system consisting of a primary and a secondary controller. The secondary controller controls motors with PI control, while the primary controller controls vehicles with type-2 FLC.

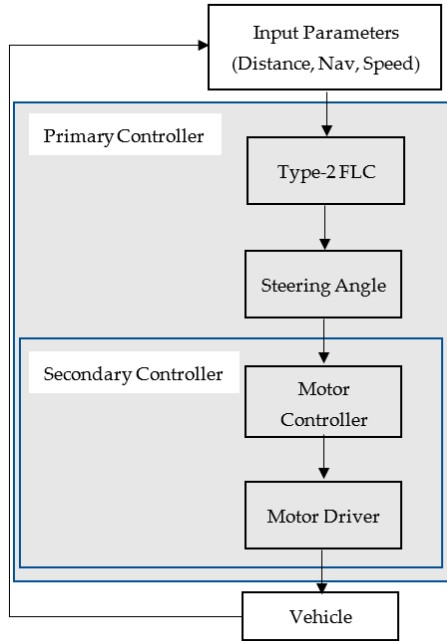

**Figure 3.** Block diagram system.

This system has a detection range of up to 10 m. The distance is divided into three categories—close, middle, and far. This car meets the criteria for a city car, with a top speed of 40 km/h. There are three speed categories—low, medium, and high. This autonomous vehicle runs according to a predetermined navigation path. The path used to benchmark

this vehicle is a set of waypoints that exist at every certain distance. There are five categories for navigation, namely turn more to the left, left, straight, right, and more to the right. The waypoint gives a signal to the car whether the car should proceed straight forward, turn, or stop.

The vehicle's position is determined by using the waypoint. In addition, the waypoint is used to know how it should move, whether straight, slightly to the right, more to the right, slightly to the left, or more to the left. This can be discovered using the side distance measuring system around the front and rear wheels. It is straight if the distance between the front and back side sensors is the same. If the front side sensor detects a distance greater than the rear side distance, this indicates to the vehicle to begin to veer to the right. If the front side sensor detects a separation that is less than the backside distance, this indicates to the vehicle to turn to the left. The difference between the front and rear side distances determines whether it will go straight, turn slightly to the right, more to the right, slightly to the left, or more to the left.

Figure 4 shows the membership function of each input and output. The maximum speed of the electric vehicle was 40 km/h. Therefore, it was set to the maximum value in the MF speed. Since the speed range was not too wide, the speed was divided into three categories—slow, medium, and fast. The MF distance classification was based on the maximum capability of the sensor used in the vehicle. The maximum detection distance of the sensor was 10 m. Since the detected distance was not too far away, the distance MF was divided into three—close, middle, and far. The vehicle's dimensions were 160 cm long and 96 cm wide. It had sensors installed in the front and rear to detect deviations from the vehicle's lane. Based on the capabilities of this sensor, navigation was divided into five categories—more to the left (big left), left, straight, right, and more to the right (big right). The output of this control system was steering wheel control, with a fixed angle according to the input. The vehicle's maximum turning angle to the right and left was 35 degrees, resulting in an encoder value of 23,000 pulses. There were five outputs—turning more to the left (big left), turning little to the left (small left), heading straight, turning little to the right (small right), and turning more to the right (big right). It was conveyed by [13] that special considerations need to overlap for determining the membership functions. This was done so that the steering smoothness factor could be achieved and passengers could be comfortable.

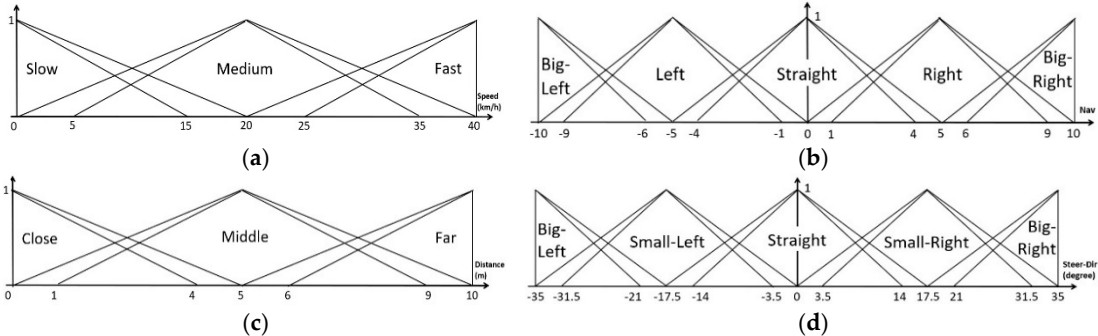

**Figure 4.** Membership function of (**a**) speed input; (**b**) navigation input; (**c**) distance input; and (**d**) steering direction output.

A combination of mathematical theories was used to obtain the necessary rule base. We adopted a mathematical technique used for calculating the possible number of occurrences in a set of items regardless of the order of selection. Any order can be selected in any combination [21]. In this study, the combination of three inputs, namely speed (three variables), navigation (five variables), and distance (three variables), amounted to 45 rules. An example of the 45 rules is: *If* the speed is slow, *And* the navigation deviates to the big left, *And* the distance is close, *Then* the steering is to the big right.

The Laboratory Virtual Electronics Workbench 2016 platform was used for all FLC computational designs. LabVIEW is a graphical computing platform that benefits from data flow-based programming, which allows it to run faster than other platforms. This software is ready to use with various hardware, including National Instruments and third-party hardware. The device used for this programming was included in the myRIO-1900 made by National Instruments. The processor used in this board was the Xilinx Z-7010.

This study has a limitation that is specific to small vehicles, particularly city cars. This vehicle had a maximum speed of 40 km/h and a detection distance of 10 m in front. Vehicles can run autonomously in special lanes marked with way-points. This study did not account for environmental disturbances, such as rain and uphill or downhill paths.

## 4. Simulation and Discussion

National Instruments' LabVIEW was used in the simulation program. The program highlighted speed, navigation, and distance visualizations. Figure 5 presents the simulation display. The left side highlighted three major indicators—speed, navigation, and distance. The maximum speed indicated by the speedometer is 40 km/h. The navigation indicator demonstrated the vehicle's direction or orientation, such as whether it was moving straight along its path, turning left, or turning right. The existing digital indicator display was also visible in the middle. When the simulation began, the autonomous indicator would initiate. The steering movement was demonstrated by turning the steering wheel to the right, straight, or left based on the input received. This was followed by a light sign indicating the vehicle's direction. The upper right corner was provided to save the recording file of all system inputs and outputs.

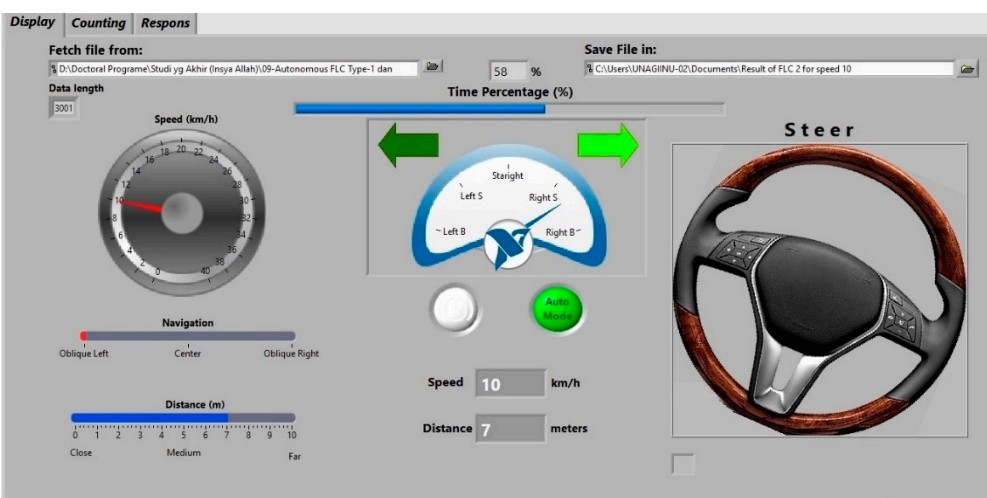

**Figure 5.** Autonomous vehicle simulation program.

Figure 6a illustrates a 10 km/h input speed over a distance of 3.5 m for the first 20 s. A sinusoidal signal was fed into the system in order for all angles to be detected. Figure 6b demonstrates the output based on the input using type-1 FLC (blue dashed line) and type-2 FLC (red dashed line) (red continuous line). The maximum steering angle obtained for type-1 fuzzy control was –35 degrees to the left and it was –31.5 degrees for type-2 fuzzy control. The minus sign in the figure represents the left direction. Type-1 fuzzy control, on the other hand, produced a rate of 35 degrees and type-2 fuzzy control produced a rate of 31.5 degrees at 15 s. Positive numbers indicate that the direction is to the right.

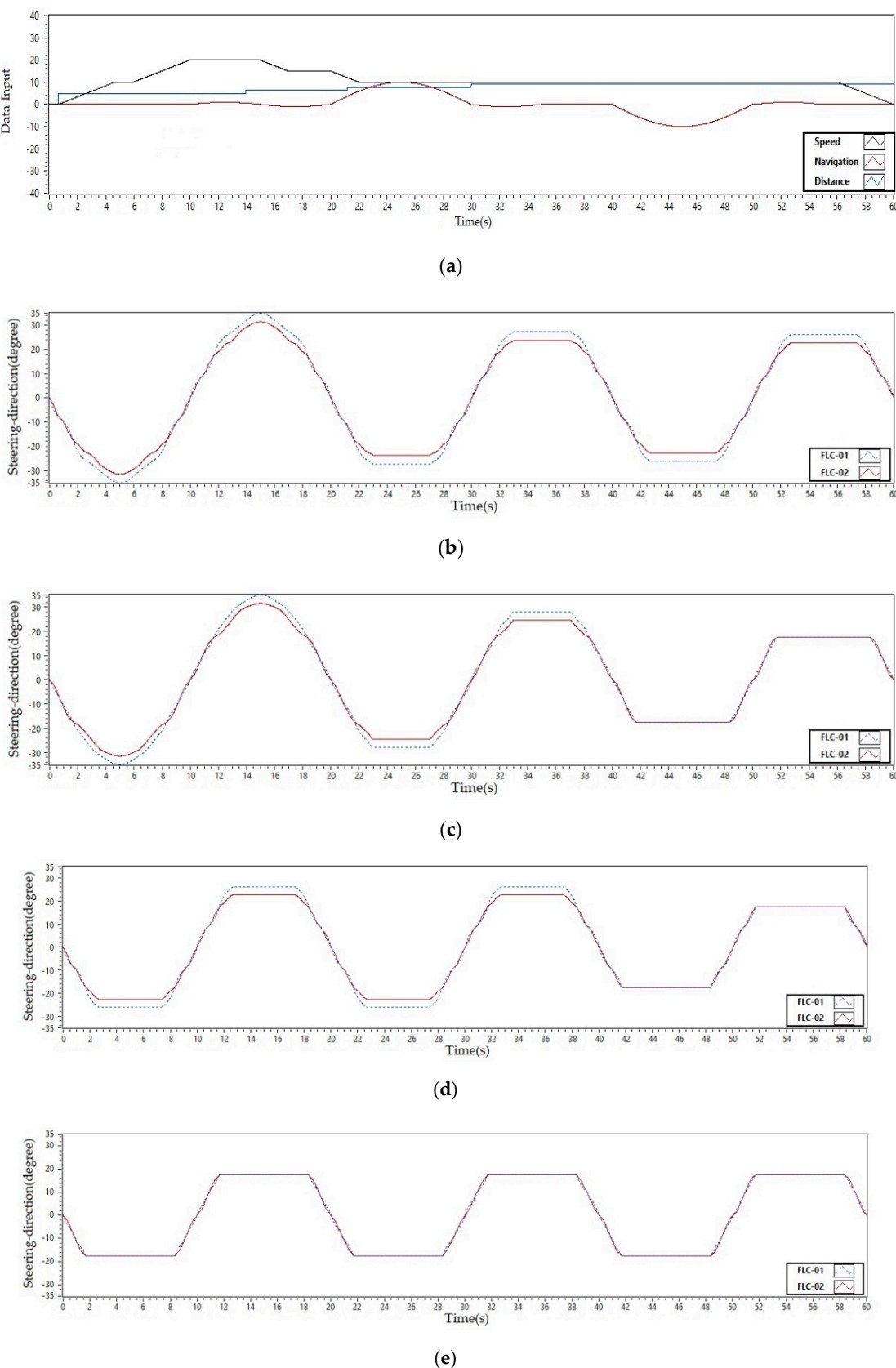

**Figure 6.** (**a**) Input of system; (**b**) system output at speed 10 km/h; (**c**) system output at speed 20 km/h; (**d**) system output at speed 30 km/h; and (**e**) system output at speed 40 km/h.

The results of the system with a seven-meter distance, at the same speed of 10 km/h and with a sinusoidal navigation signal input form, are presented in Figure 6b. The time interval ranged from 20 to 40 s. Different results were obtained from the previous settings with this combination of inputs. At a starting time of 22.980 s, the output for the type-1 FLC reached a steering angle of –27.22 degrees. The minus sign indicates the left direction. The steering angle of the type-2 FLC was –23.587 degrees. This value lasted for a maximum of 27.060 s.

In the same figure, the distance detected was 10 m in the third part, which lasted 40 s to 60 s. This system was configured with a speed of 10 km/h and a sinusoidal navigation signal. The maximum steering angle obtained in 42.740 s was 26.250 degrees to the left for type-1 FLC and 22.750 degrees to the right for type-2 FLC. This state lasted up to 47.340 s. With the same input distance, in 52.740 s, the steering control to the right was generated at 26.250 degrees for type-1 FLC and at 22.750 degrees for type-2 FLC. This output value lasted up to a maximum of 57.340 s.

Figure 6b shows that at a speed of 10 km/h, there were several variations of the steering angle output. With a relatively close distance, the resulting steering angle was of great value so that the car could avoid existing obstacles. With type-2 FLC, the resulting steering angle was high, but not until it reached the maximum. This was different from the results of the type-1 FLC output. Therefore, it can be said that the type-2 FLC produced a smoother steering system than the type-1 FLC. At moderate distances, the control automatically reduced the steering angle. Similarly, if the distance between the car and the barrier was far, the resulting steering angle would be smaller as well. This shows that at low speeds and with close obstacles, the control produced a small control angle. This was done for the sake of convenience. To summarize, the type-2 FLC method produced smoother results than the type-1 FLC method.

For the following input, the speed increased to 20 km/h, with sinusoidal for navigation and a distance of 3 m, resulting in an increased steering angle response to a peak of –35 degrees for type-1 FLC and –31.5 degrees for type-2. The negative value in Figure 6c indicates that the direction was to the left. The peak was reached in 5 s. On the other hand, when the sine input ran in the opposite direction, the steering system output reached its peak in 15 s, with a value of 35 degrees for type-1 FLC and 31.5 degrees for type-2 FLC.

At a distance of 7 m, the next position caused a different response from the previous settings (medium distance of seven meters in 20 s to 40 s). Although the navigation input was sinusoidal, the output from 22.980 s to 27.060 s showed a similar value. The resulting values were –28.00 degrees for type-1 FLC and –24.553 degrees for type-2 FLC. The positive value was obtained between 32.980 s and 37.06 s for the next half of the sinusoidal navigation input condition.

A time interval of 40 to 60 s indicates an input distance of 10 m. The response obtained was that the steering system moved according to the input until the time was 41.720 s, reaching a value of –17.5 degrees for type-1 and type-2 FLC. This value lasted up to 48.360 s. In the sinusoidal navigation input condition, the value of 17.5 degrees (both type-1 FLC and type-2 FLC) was obtained when it reached a time of 51.720 s to 58.360 s.

According to Figure 6c and the description above, at a speed of 20 km/h with a sinusoidal variation of the navigation input, the short distance produced by both types was similar to that produced at a low speed (10 km/h). For a medium distance (seven meters), the resulting control was slightly different between the two FLCs, but the type-2 FLC had a smaller steering angle. However, these two controls showed the same results at a far distance (10 m). The results show that the type-2 FLC was more capable than the type-1 FLC at providing a smoother steering system.

In the following simulation experiment, the input speed was 30 km/h and the navigation was sinusoidal, with variations in the distance used. The close-range distance was 3.5 m at 0 to 20 s, the medium distance was 7 m at 20 to 40 s, and the far distance was 10 m at 40 to 60 s. The results are presented in Figure 6d. The resulting steering angle response was –26.25 degrees for type-1 FLC and –22.750 degrees for type-2 FLC at the close-range

input. It took between 2.7 s and 7.3 s. The duplicate value in the opposite direction occurred at 12.7 s to 17.3 s.

For a medium distance of seven meters, the resulting response at 22.720 s to 27.32 s showed a maximum value of –26.25 degrees for type-1 FLC and –22.750 for type-2 FLC. The positive direction was achieved in 32.720 s to 37.32 s.

The response obtained when the input was a relatively far distance of 10 m was 17.5 degrees for both types of control. This occurred between 41.720 s and 48.360 tseconds. The opposite value, which is 17.5 degrees, was obtained in 51.720 s to 58.36 s.

According to the previous description and Figure 6d, the steering angle control value at 30 speeds for short and medium distances was similar. Type-1 FLC had an angle of 26.25 degrees, while type-2 FLC was at 22.750 degrees. The steering angle of type-2 FLC was smaller than that of type-1 FLC. This indicates that type-2 FLC had better control at high speeds. Furthermore, the shape of the graphic response was also smoother. Meanwhile, with a long-distance input of 10 m, both showed the same steering angle results.

At a high speed (according to city car specifications) of 40 km/h and at a distance of 3.5 m, both type-1 and type-2 FLC achieved a maximum steering angle of –17.5 degrees in 1.68 s to 8.32 s. The direction shown was to the left because it was negative. Meanwhile, at 11.68 s to 18.32 s, the steering angle showed a value of 17.5 degrees in positive numbers, indicating the right direction.

The same thing happened when the obstacle distance was increased to seven meters. Both the right and left steering directions still showed a value of 17.5 degrees in response to the sinusoidal navigation input. The steering angle of the autonomous vehicle remained at 17.5 degrees both to the right and to the left for a 10-meter distance. The control results are presented in Figure 6e.

The steering system produces a small steering angle at high speeds, according to the analysis. This prevents uncontrollable movements in the vehicle caused by a large steering angle when going fast. According to Figure 6e, the type-2 FLC (dotted line) output was smoother than the type-1 FLC output (dashed line). This indicates that the type-2 FLC had better high-speed control.

## 5. Experimental and Discussion

Following the simulation, we applied it to the Sultan Agung Autonomous Vehicle (SAAV). The SAAV is an electric car designed by H-Molex, with dimensions of 160 cm long and 96 cm wide. Figure 7 shows a SAAV equipped with an autonomous system and steering motor. A modified Daihatsu Xenia 2012 steering system, consisting of an EPS (electric power steering) motor and its column, was used as the driving force for the steering wheel. The motor has a specification of 2000 rpm 12 volts. The system steering uses a rack and pinion system, the turning angle of which is 35 degrees with an encoder value of 23,000 pulses.

To control the motor, conventional control was used, which was proportional–integral and differential. The result of type-2 FLC, namely the steering angle, became the input of this control. This control was chosen because of its simplicity and ease of obtaining the parameters. There were several methods used to obtain the control parameter values. One was the Ziegler–Nichols method [30,31]. The Ziegler–Nichols method chosen was the closed-loop type, often known as type-2. Its equations are shown in Table 1. The step signal was converted into an input for the plant, and the plant's response was observed. To generate oscillations, the gain was continuously added until the output reached a steady-state value to get oscillations. The gain value (K) obtained was 6, with an oscillation time (Pu) of 0.0064 s. Table 2 shows the values obtained by the type-2 Ziegler–Nichols method.

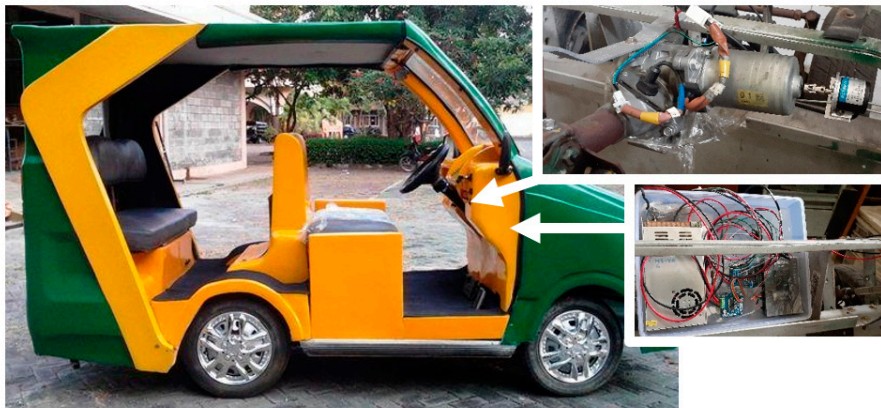

**Figure 7.** SAAV, EPS motor, and control system.

**Table 1.** Control parameters using Ziegler Nichols method.

| Control | Kp | Ti | Td |
| --- | --- | --- | --- |
| PI | 0.45 K | Pu/1.2 | 0 |
| PD | 0.8 K | 0 | Pu/8 |
| PID | 0.6 K | Pu/2 | Pu/8 |

**Table 2.** Control parameters for secondary controller.

| Control | Kp | Ti | Td |
| --- | --- | --- | --- |
| PI | 2.7 | 0.0053 | 0 |
| PD | 4.8 | 0 | 0.0008 |
| PID | 3.6 | 0.0032 | 0.0008 |

Figure 8 shows the motor control program. The program consisted of inputs for the motor control set-points, equipped with PID control inputs, namely Kp, Ti, and Td. The control output produced the PWM value. This value directs the motor to the right or vice versa according to the control. The encoder section was used to determine the results of the motor rotation generated by the rotary encoder. The encoder value output is useful for providing predefined control feedback. The bottom side shows that all data (input and output) were stored in an array, and the shift register was used to store the results in a file, so that they could, then, be analyzed.

Figure 9a shows the output of the motor when amplification was applied; the value continuously increased until oscillation. These results occurred when K = 6 was strengthened, and the oscillation period was 0.0064 s. After obtaining these two values, the motor was controlled using the PI, PD, and PID methods, as shown in Table 2. For PI control with values of Kp = 2.7 and Ti = 0.0053, the response was as shown in Figure 9b; PD control with values of Kp = 4.8 and Td = 0.0008 was as shown in Figure 9c; for PID control with Kp = 3.6, Ti = 0.0032, and Td = 0.0008, the response was as shown in Figure 9d. The PID control achieved the settling time at 14.6 s, the PD control achieved it at 10.8 s, and the PI control achieved it at 9 s. Based on these results, the PI control was better than the PID and PD control. Therefore, the PI control was used in the SAAV EPS motor. This is a secondary control that is responsible for executing the number of steering rotations.

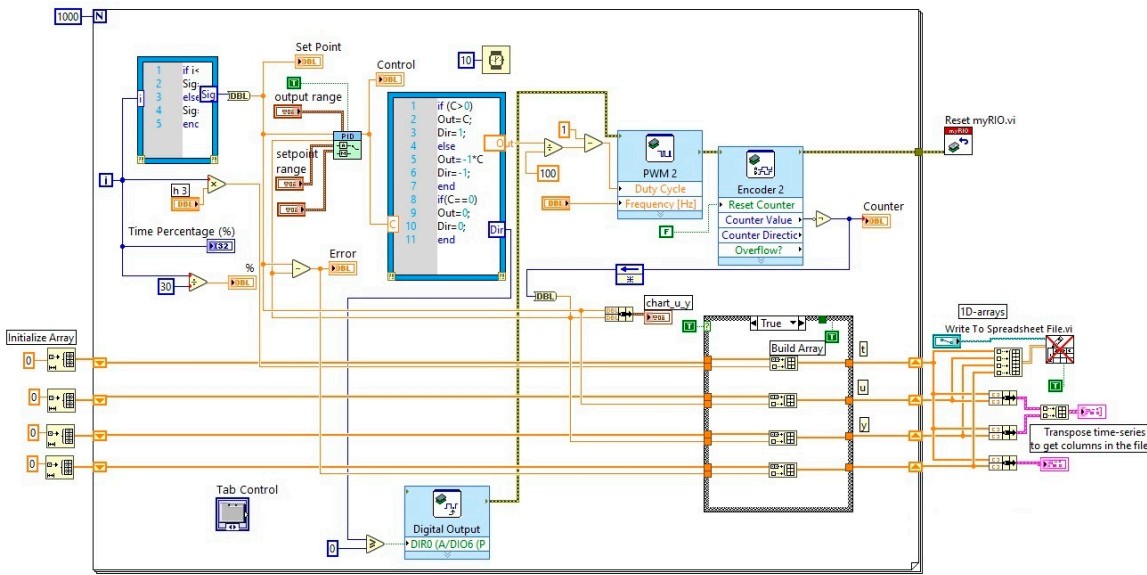

**Figure 8.** Program of motor control system.

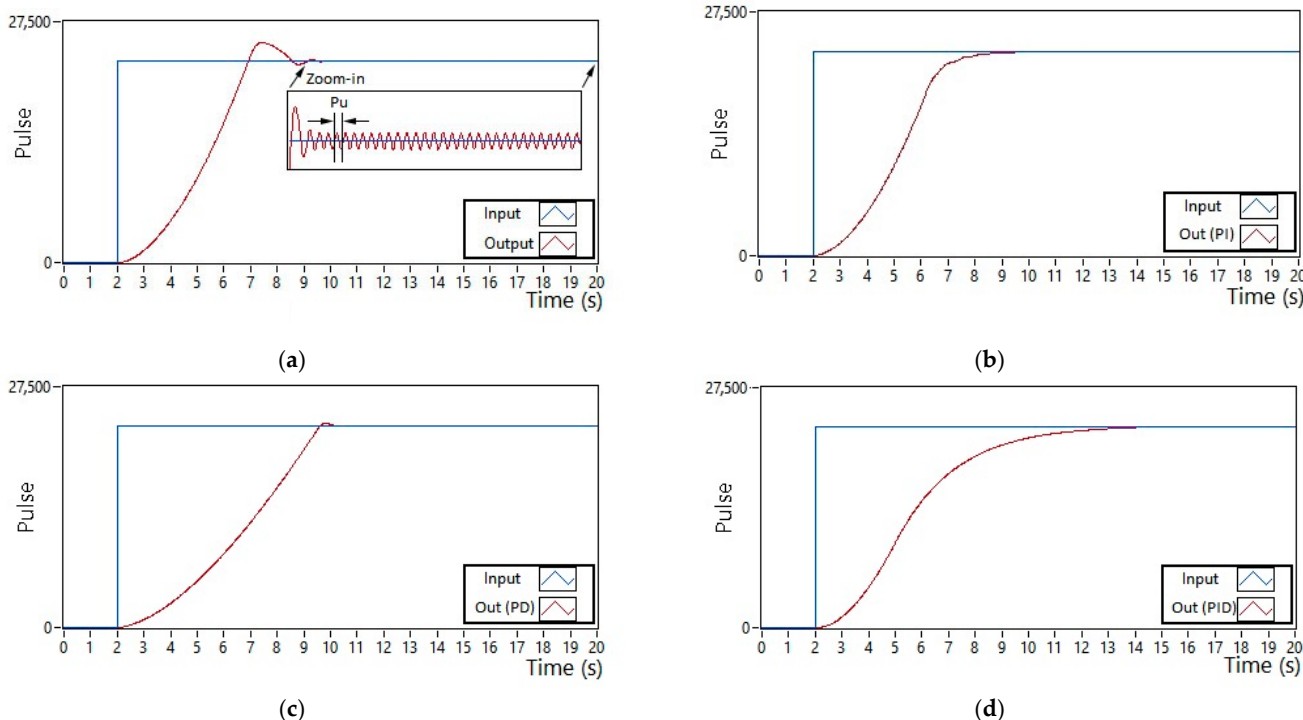

**Figure 9.** Result of motor control system. (**a**) Oscillation output; (**b**) PI; (**c**) PD; and (**d**) PID.

A trial of several positions of the EPS motor was carried out to test the secondary control. Following the design, the steering angle of this vehicle was 35 degrees for both the left and right directions, which was converted to 23,000 pulses. There were five outputs—turning more to the left (big left = −23,000 pulses), turning little to the left (small left = −12,000 pulses), heading straight (0 pulses), turning little to the right (small right = 12,000 pulses), and turning more to the right (big right = 23,000 pulses). An up-down-up signal was used to test the position control at these angles. Figure 10 shows the output signal (PI control = red continuous line; PD control = blue continuous line; PID control = green continuous line) in response to the input signal (blue dashed line). The initial state was an input of 23,000 pulses. The PI control achieved the settling time at

1.08 s, the PD control achieved the settling time at 1.44 s, and the PID control achieved it at 1.52 s. It can be seen that the PD control overshot until it reached a value of 23,834 pulses and oscillated. The pulse was lowered to 12,000 to change the motor position. The PI control achieved the settling time at 5.4 s, the PD control achieved it at 5.78 s, and the PID control achieved it at 5.86 s. In this condition, the PD control reached an overshoot of 11,336 pulses. For the straight position, the PI, PD, and PID controls reached the settling time at 10.6 s, 10.78 s, and 11.18 s, respectively. The PI control reached the settling time at 15.86 s, the PD control at 16.22 s, and the PID control at 16.48 s when the input signal was lowered to –23,000 pulses. The PD control in this state overshot up to –23,265 pulses and oscillated. The input pulse was increased to –12,000 pulses; this caused the PI, PD, and PID controls to reach the settling time at 20.6 s, 20.92 s, and 21.14 s, respectively. This position change caused the PD control overshoot to reach –12,401 pulses. The next straight position caused the PI control to reach the settling time at 25.64 s, the PD control at 25.9 s, and the PID control at 26.16 s. As before, the PD control oscillated and overshot at 313 pulses. A similar condition also occurred in the next position change. Based on Figure 10 and the data obtained, it can be seen that this position change can be handled properly by the PI control. The PD control with overshoot and oscillation values, and the PID control with the slowest settling time are described later.

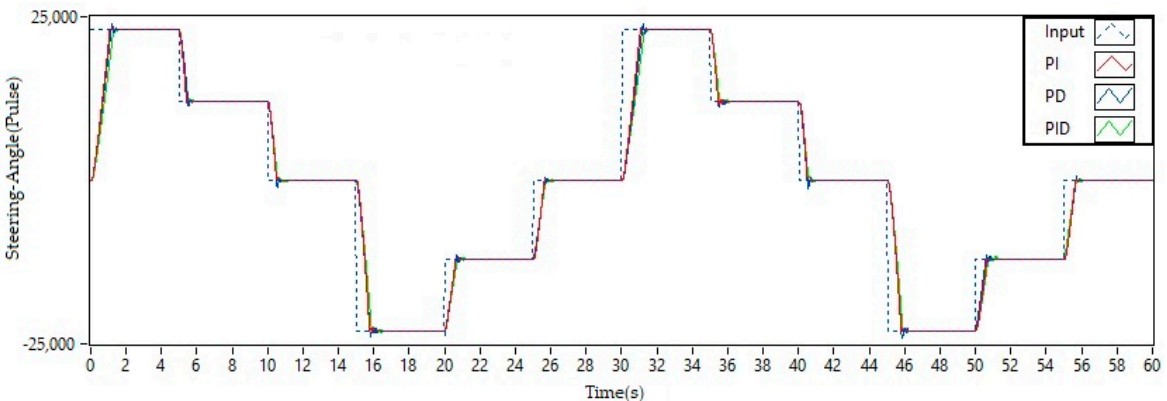

**Figure 10.** Position control response to the up-down-up input signal.

Following the discovery of good motor control, namely PI control, the next step was to provide the plant with the same input signal as the simulation. Figure 11a shows the simulation (blue dashed line) and reality (red continuous line) results for type-1 FLC (the left-hand side) and type-2 FLC (the right-hand side) at a speed of 10 km/h. At a distance of 3.5 m, it appears that the control of the type-1 FLC steering wheel movement was not very similar to the simulation. In contrast to the type-1 FLC, the movement of the simulated steering wheel and the real one was more similar in type-2 FLC. Type-1 FLC reached a simulation value of –22,995 pulses and a reality value of –21,176 pulses in the left direction. The value in type-2 FLC simulation was –20,376, whereas its reality value was –19,641 pulses. The minus sign in the figure indicates that the direction was to the left. For type-1 FLC, the difference between simulation and reality was 1819, and for type-2 FLC, it was 726. For the right direction, type-1 FLC reached a value of 22,995 in the simulation and 21,539 in reality, whereas the type-2 FLC achieved a value of 20,367 in the simulation and 19,639 in reality. The differences in the timing between the simulation and reality for type-1 and type-2 FLC were 1456 pulses and 728 pulses, respectively.

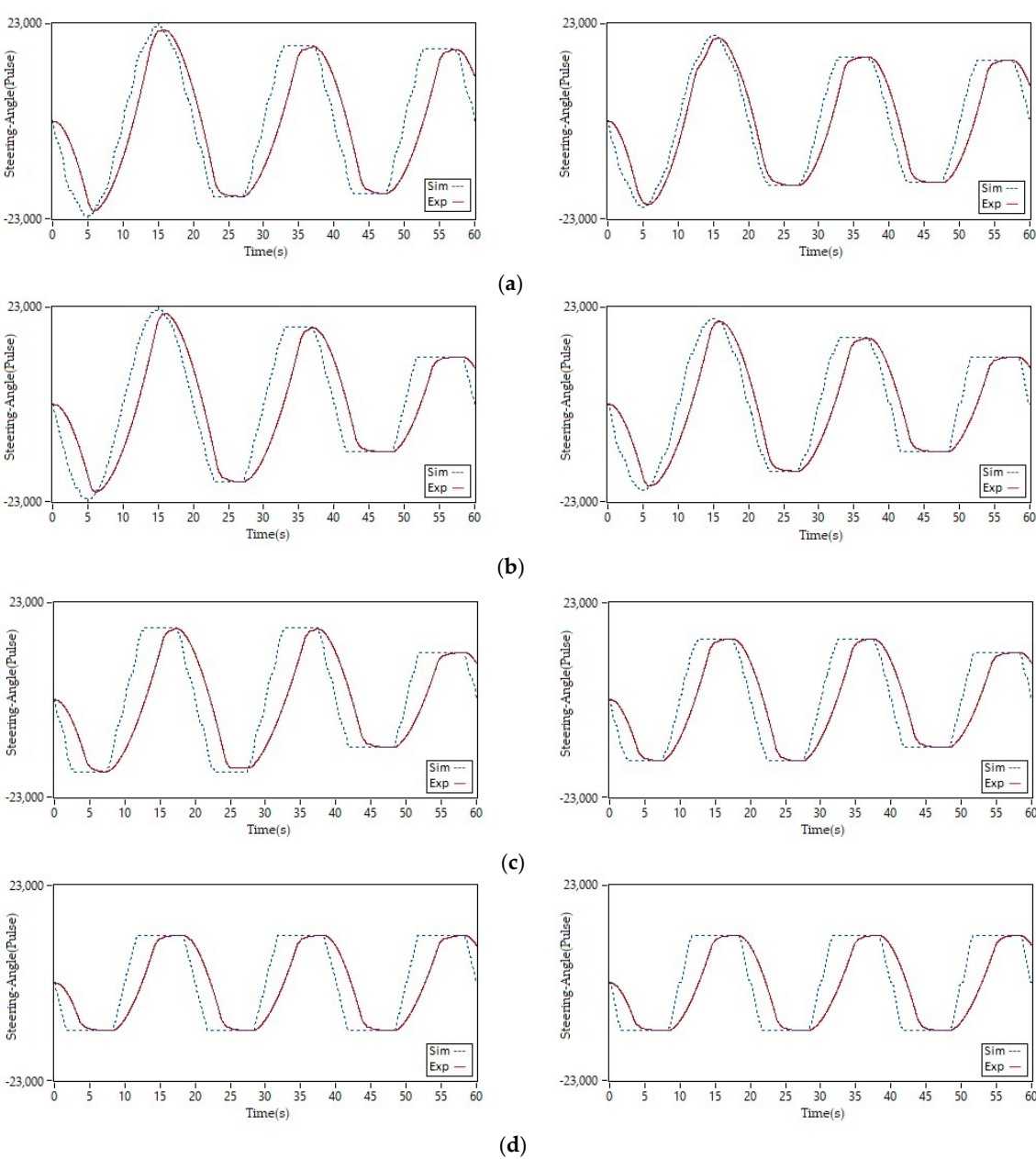

**Figure 11.** Comparison between simulation and experiment output signal generated at speeds of (**a**) 10 km/h; (**b**) 20 km/h; (**c**) 30 km/h; and (**d**) 40 km/h.

At a distance of seven meters at the beginning, both controls produced the same response, but the type-2 FLC was more stable at the peak of the turn. In the left direction, type-1 FLC reached a value of −17,739 pulses for the simulation value and a value of −17,731 pulses for the reality value. The value in the type-2 FLC simulation was −15,111, whereas the reality value was −15,108 pulses. The difference between the simulation and reality for type-1 FLC was eight pulses and for type-2 FLC, it was three pulses. For the right direction, type-1 FLC reached a simulation value of 17,739 and a reality value of 17,598, while the type-2 FLC reached a value of 15,111 in the simulation and 15,097 in reality. The differences in the pulses between the simulation and reality for the type-1 and type-2 FLC were 141 and 14, respectively. At a distance of 10 m, both controls produced the same response. Type-1 FLC reached a simulation value of −17,082 pulses and a real value of −17,062 pulses in the left direction. The value in the type-2 FLC simulation was −14,454, whereas the real value was −14,450 pulses. The difference between the simulation and reality for type-1 FLC was 20 pulses and for type-2 FLC, it was 4 pulses. Type-1 FLC

achieved values of 17,082 in the simulation and 16,959 in reality for the right direction. Meanwhile, type-2 FLC reached values of 14,454 in the simulation and 14,431 in reality. The difference in the timing between the type-1 and type-2 FLC simulation and reality was 123 pulses and 23 pulses, respectively. Based on Figure 11a and Table 3, the type-2 FLC in all simulations and experiments had nearly the same form of control. Based on the value differences between the two, different things happened to type-1 FLC because the simulation and experiment were somewhat different.

**Table 3.** Comparison between simulation and experiment output signal generated at a speed of 10 up to 40 km/h. (Dir = direction; Sim = simulation; Exp = Experiment).

| Distance(m) | | | 3.5 | | 7 | | 10 | |
|---|---|---|---|---|---|---|---|---|
| Speed (km/h) | FLC | Dir | Sim | Exp | Sim | Exp | Sim | Exp |
| 10 | Type-1 | Left | −22,995 | −21,176 | −17,739 | −17,731 | −17,082 | −17,062 |
| | | Right | 22,995 | 21,539 | 17,739 | 17,598 | 17,082 | 16,959 |
| | Type-2 | Left | −20,367 | −19,641 | −15,111 | −15,108 | −14,454 | −14,450 |
| | | Right | 20,367 | 19,639 | 15,111 | 15,097 | 14,454 | 14,431 |
| 20 | Type-1 | Left | −22,995 | −20,691 | −18,396 | −18,376 | −11,169 | −11,169 |
| | | Right | 22,995 | 21,469 | 18,396 | 18,170 | 11,169 | 11,166 |
| | Type-2 | Left | −20,367 | −19,242 | −15,768 | −15,764 | −11,169 | −11,169 |
| | | Right | 20,367 | 19,604 | 15,768 | 15,702 | 11,169 | 11,163 |
| 30 | Type-1 | Left | −17,082 | −17,044 | −17,082 | −16,185 | −11,169 | −11,169 |
| | | Right | 17,082 | 16,906 | 17,082 | 16,803 | 11,169 | 11,162 |
| | Type-2 | Left | −14,454 | −14,443 | −14,454 | −14,441 | −11,169 | −11,169 |
| | | Right | 14,454 | 14,430 | 14,454 | 14,424 | 11,169 | 11,164 |
| 40 | Type-1 | Left | −11,169 | −11,169 | −11,169 | −11,169 | −11,169 | −11,166 |
| | | Right | 11,169 | 11,164 | 11,169 | 11,163 | 11,169 | 11,155 |
| | Type-2 | Left | −11,169 | −11,168 | −11,169 | −11,169 | −11,169 | −11,169 |
| | | Right | 11,169 | 11,160 | 11,169 | 11,164 | 11,169 | 11,158 |

Figure 11b shows a comparison of the results between the type-1 FLC (the left-hand side) and type-2 FLC (the right-hand side) simulation and reality at a speed of 20 km/h. There was a slight difference between the left and right for type-1 and type-2 FLC, respectively. Based on the value difference, type-2 FLC outperformed type-1 FLC despite having nearly the same shape. Figure 11c shows that at a speed of 30 km/h for a distance of 3.5 m, Type-2 was better than the type-1 FLC, while other distances show the same shape. Figure 11d, shows the same shape for a speed of 40 km/h, although there is a slight difference in the pulses generated. Table 3 summarizes the differences in the values achieved for the steering wheel between the type-1 FLC and type-2 FLC.

Figure 12 shows the FLC system input for the SAAV. In the first 10 s, there was a vehicle-steering disturbance. Based on the type-1 FLC output, this disturbance was first responded to at 10.46 s, with an encoder value of 657 pointing in the right direction. The encoder value increased to 1314 until it peaked at 10.48 s by producing an encoder value of 1971 in the same direction, and it began to slope until 14.62 s. This disturbance began to be responded to by type-2 FLC at 10.88 s, with a value of 657 pulses in the right direction. The value of 657 remained constant until the peak of 12.46 changed to 1314 and returned to 657 at 14.24 s. It was discovered that the type-2 FLC output responded to this disturbance in a shorter time and produced a smaller encoder pulse value. This means that the turning angle caused was smaller than the type-1 FLC output.

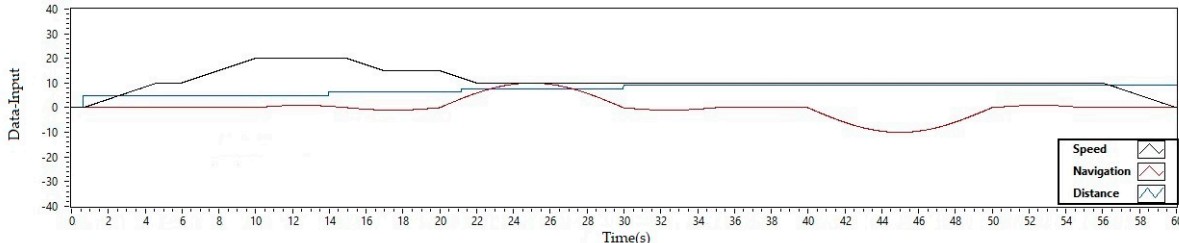

**Figure 12.** Input of system.

The type-1 and type-2 FLC outputs are shown in Figure 13a,b, respectively. When the vehicle started turning right at 20 s, type-1 FLC responded with an initial value of 657 pulses at 20.06 s. It continued to increase until 22.68 s with an encoder value of 17,082 pulses. This type-1 FLC output lasted until 27.28 s and gradually decreased until 29.92 s. Meanwhile, type-2 FLC started to respond with a value of 657 pulses at 20.12 s. This value continued to increase to 14,454 until 22.46 s. This value began to decrease at 27.52 s and ended at 29.88 s. These results show that the turning angle produced by type-2 FLC was smaller and smoother than that produced by type-1 FLC.

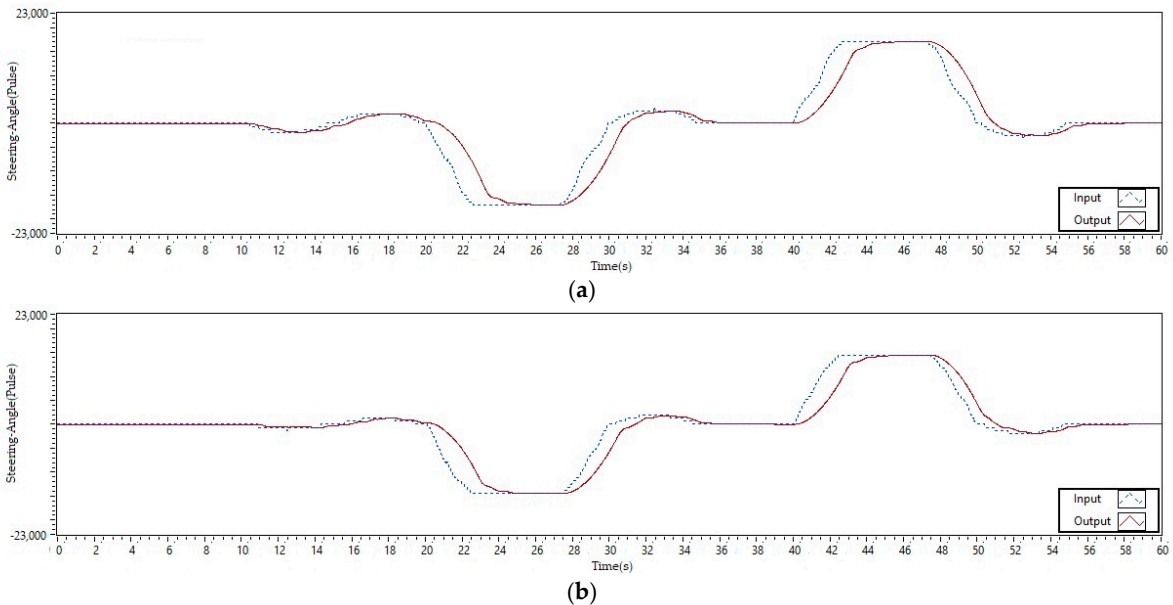

**Figure 13.** System output (**a**) using type-1 FLC and PI control and (**b**) using type-2 FLC and PI control.

Another disturbance occurred when the vehicle resumed its straight path after passing through the previous turn. It occurred at 29.98 s and was responded to by type-1 FLC at 30.22 s with a value of 657 pulses. This disturbance reached its peak at 32.4 s with a response of 3285 pulses. After that, the value decreased to 34.7 s. The type-2 FLC response to this disturbance resulted in 657 pulses occurring at the time of 30.38 s. The maximum control value was 1971 pulses from 31.62 s to 33.3 s. This value gradually dropped until it reached 34.54 s. Based on these results, it was discovered that when subjected to the same disturbance, type-2 FLC responded with smaller values and a shorter time span.

The vehicle began to turn left at 40 s. At 40.04 s, type-1 FLC responded with a value of 657 pulses. This value increased until it reached a peak of 17,082 from 42.62 s to 47.34 s. This value gradually decreased until it returned to its original state at 49.92 s. Type-2 FLC produced a value of 657 pulses and gradually increased to a peak of 14,454 pulses at 42.46 s. This lasted until 47.5 s and decreased continuously to its original position at 49.88 s. Therefore, it can be said that the type-2 FLC output produced smoother turning angle control and a shorter time span compared to type-1 FLC.

Figure 13 shows that the DC motor on the EPS could be properly controlled using PI control. These results show that the secondary control processed the FLC output (dashed line) to drive the motor (continuous line). As a result, the motor response could closely follow the FLC results. There was a time delay at 20 s when the vehicle turned right and returned to its original direction at 27 s. However, the error that occurred at that time was very minor. It determined that the motor response directly followed the FLC output because the delay was still tolerable. At 40 s, the same thing happened when the vehicle began to turn left and returned to its original direction at around 50 s. The delay between the FLC output and the motor response was also very slight. Therefore, with PI control, the motor could respond well to the incoming input and produce fast and precise control in response to the FLC results.

Since type-2 FLC showed better results, the following test used PD and PID controls for type-2 FLC. The test results with PD control are shown in Figure 14a. The test results with PID control are shown in Figure 14b. When the vehicle began to turn right at 20 s, the PD motor control began to respond at 21.32 s. For PID control, this movement was responded to at 20.34 s. The PD control took longer to respond than the PID control. On the other hand, the PD control quickly reached the specified turning angle at 25.76 s. In contrast, PID control reached its maximum angle at 27.74 s.

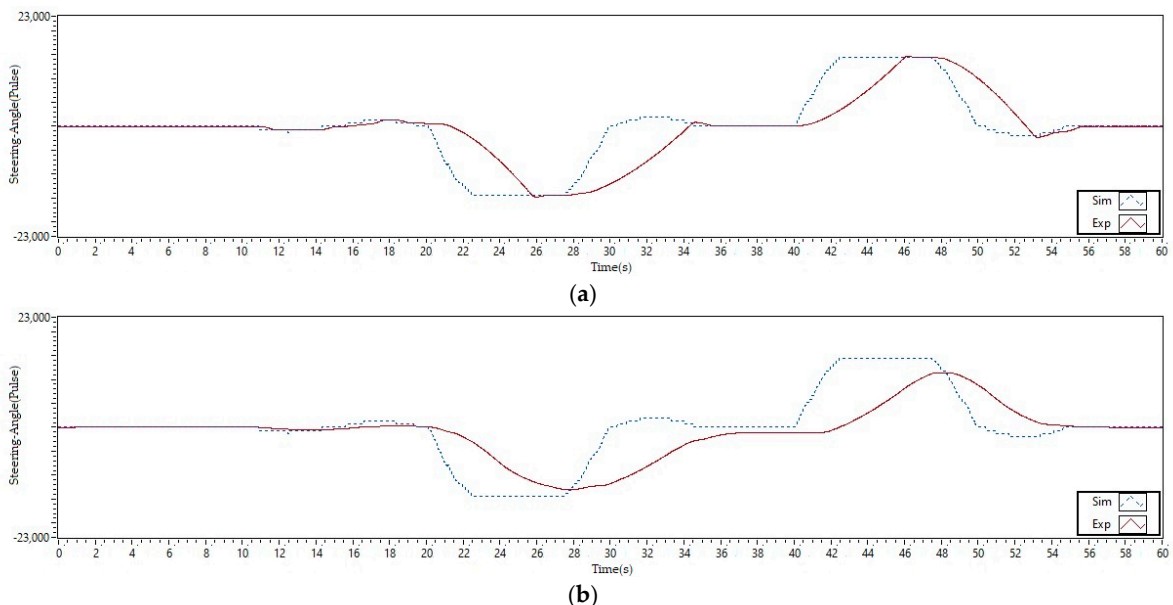

**Figure 14.** System output (**a**) using type-2 FLC and PD control (**b**) using type-2 FLC and PID control.

The PD control experienced a steady state according to the input at 26.72 s to 27.56 s, whereas the PID control never reached a steady state. The steering system with the PD control returned to its original state at 36.18 s, while the PID control system returned in 37.54 s. For a 40-second left turn, the PD control responded faster at 40.14 s, while the PID control responded at 42.46 s. At 46.42 s, the PD control reached a steady-state value according to the input and continued until 47.68 s. Similar to the first turn, the PID control switched to the opposite state without experiencing a steady state at 48.24 s. While in PD control, the steering wheel rotated in the opposite direction according to the input at 47.68 s and returned to a straight state at 55.64 s. Straight steering was achieved with PID control at 56.96 s. Based on these results, it can be seen that PID control responded 0.98 s faster than the PD control at the beginning. However, the PID control was unable to maintain this value, never reaching the specified value. The PD control reached and maintained the expected value for 0.84 s. In the second corner, the PID control was unable to respond as quickly as it had previously. A superior PD control response resulted in a 2.32-second difference.

The PD control reached steady-state in 1.26 s, whereas the PID control never did. The PD control was 1.32 s faster than the PID control to achieve straight steering. Figure 14a,b shows a comparison of the performance of the methods used in this study. When small disturbances occurred for the first time at 10 s, the PID control could handle them better than the other controls. However, this was not always the case because the steering wheel remained straight due to the slow PID response. It was demonstrated that the PID control was the last to respond when the vehicle turned right or left.

The PI control responded quickly from 20.5 s to 24.64 s by generating 13,793 pulses. The steering wheel moved slowly until it reached 26.22 s. The steering wheel began to stop moving while maintaining a turn for 1.44 s. PD control was a bit slow to respond, taking 21.32 s to 25.76 s with 14,780 pulses. The steering wheel stopped moving for 0.84 s. PID control responded in 20.34 s to 27.74 s and immediately reversed directions. It made the passengers feel uncomfortable because these controls never stopped moving. In the second turn, which is to the left, the PI control generated 13,002 pulses in 40.12 s to 43.98 s. The steering wheel turned slowly until 46.4 s and then stopped turning for 1.26 s. While the PD control began to respond at 40.14 s and ended at 46.02 s, it produced 14,426 pulses. PID control responded in 42.46 s to 48.24 s. Without pausing to move, the steering wheel immediately reversed direction. Based on the values obtained, PI control could provide comfort. It was indicated by a quick response to reach the expected value and a slow response when almost there. Following this, PI control could maintain a longer steady-state value compared to the other controls. PD control had a slower response than PI control PD control reached the desired value without decelerating. Of course, this affected the passenger comfort. PID control is not recommended because it cannot guarantee comfort or accuracy in steering control, and it never achieved the desired value. The steering wheel also changes direction immediately. In comparison to the others, the level of comfort of PID control comes in last.

Figure 15 shows the steering output based on the variation of the loading on the motor shaft. Loading was applied to the EPS motor shaft to test the performance robustness. These loads came in a variety of weights, including 1 kg, 2 kg, 3 kg, 4 kg, and 5 kg. The proposed control provided input to the EPS motor, allowing it to move to the right or left. The results of this test are shown in Figure 14. The motor generated 19,700 pulses with no load. It can be seen that at a load of 1 kg, the motor still moved well and achieved a value of 19,658 pulses. For a load of 2 kg, the motor could also move following the input by reaching a value of 19,623 pulses. When loaded with 3 kg, the motor could only produce 1500 pulses. At this load, it appears that the motor lost its ability to produce the required output. When the motor was loaded with 4 kg, the output was 1000 pulses; when loaded with 5 kg, the output was 500 pulses. Based on the results, it can be stated that the motor could only be loaded up to 2 kg and that any load greater resulted in a significant decrease in performance.

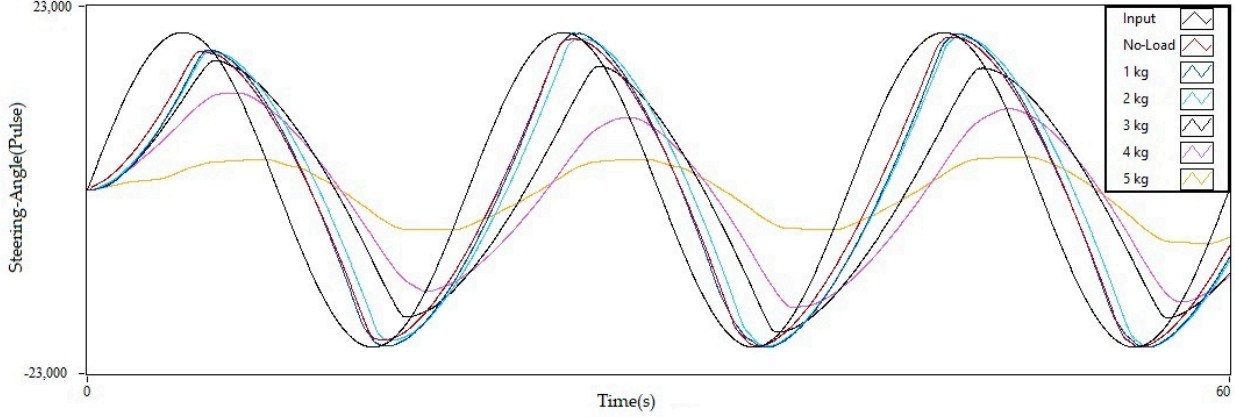

**Figure 15.** Pulse output on the steering with loading variation of the motor shaft.

## 6. Conclusions

This study used two types of controls—FLC and conventional control. The primary control was based on fuzzy logic, while the secondary control was based on conventional PI control. Both of these controls were applied to a SAAV. A SAAV is an electric car with a city car design. There were three inputs to this fuzzy control—speed, navigation, and distance. Speed and distance were divided into three membership functions, while navigation was divided into five membership functions. The rule set used had 45 rules. The results show that at a low speed (10 km/h) and medium speeds (20 km/h and 30 km/h), type-2 FLC was superior to type-1 FLC. At a high speed (40 km/h), the type-2 FLC control and the type-1 control showed the same results. However, type-2 FLC showed smoother results than type-1 FLC. The results of this fuzzy control were used as input for the secondary control, namely conventional control. The results obtained indicate that the PI control was better than the PD and PID control. The time differences between the steady-state and settling time were 1.8 s and 5.6 s, respectively. This is significant for autonomous vehicle driving. Based on the data obtained, it can be concluded that the type-2 FLC and conventional PI control are more capable of controlling autonomous vehicles.

In the future, this study can be expanded on to include a broader range of fuzzy input. Starting with long-range detection, it can also detect the vehicle's side and, if possible, the rear. As a result, vehicle safety could be guaranteed because all sides of the vehicle would be detected and turned right or left. Better methods, such as genetic algorithms, can be used to determine the optimal fuzzy values.

**Author Contributions:** Conceptualization, B.A.; methodology, B.A.; software, B.A.; validation, B.A., Z.N., B.Y.S. and S.A.D.P.; formal analysis, B.A.; investigation, B.A., B.Y.S. and S.A.D.P.; resources, B.A.; data curation, B.A.; writing—original draft preparation, B.A.; writing—review and editing, B.A., B.Y.S. and S.A.D.P.; visualization, B.A., B.Y.S. and S.A.D.P.; supervision, Z.N., B.Y.S. and S.A.D.P.; funding acquisition, B.A. All authors have read and agreed to the published version of the manuscript.

**Funding:** This research received no external funding.

**Acknowledgments:** Electrical Department of Universitas Sriwijaya and Universitas Islam Sultan Agung.

**Conflicts of Interest:** The authors declare no conflict of interest.

## Glossary

| | |
|---|---|
| $\delta_f$ | front wheel steering angle [rad] |
| $\delta_r$ | rear wheel steering angle [rad] |
| $C$ | c.g. = center of gravity |
| $L$ | total wheel base ($l_f + l_r$) [m] |
| $l_f$ | longitudinal distance from c.g. to front tires [m] |
| $l_r$ | longitudinal distance from c.g. to rear tires [m] |
| $\beta$ | slip angle at vehicle c.g. [rad] |
| $R$ | turn radius of the vehicle or radius of road [m] |
| $\psi$ | yaw rate of vehicle [rad/s] |
| $a$ | inertial acceleration [$\text{ms}^{-2}$] |
| $F_{yf}$ | the lateral tire force on front tires [$\text{kg m s}^{-2}$] |
| $F_{yr}$ | the lateral tire force on rear tires [$\text{kg m s}^{-2}$] |
| $C_a$ | cornering stiffness of tire [$\text{kN rad}^{-1}$] |

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
