# Peer review of "Steering Control in Electric Power Steering Autonomous Vehicle Using Type-2 Fuzzy Logic Control and PI Control"

_wevj, doi:10.3390/wevj13030053_

Round 1
Reviewer 1 Report
The state-space model on line 167 is incorrect. No state is shown on the right-hand side of the model.
It is not clear why and how the membership functions in Fig. 4 are chosen. The motivation for the rules on lines 225-234 is not clear.
It has to show that the proposed controller can yield robust performance of the system when some torque disturbance is imposed on the motor shaft. In addition, it needs to give the extend of the torque disturbance that the system can tolerate.
To control the SAAV steering system is the main theme of the manuscript. In this regard, the simulation section of the proposed fuzzy controller should be combined with the experimental section. The descriptions in the simulation and experimental sections should be reorganized clearly.
Reviewer 2 Report
This study proposed a steering control approach for autonomous vehicles based Type-2 fuzzy logic control and PI. There are some suggestions for the authors to improve the paper.
(1) Firstly, the authors should state clearly why the fuzzy control based method was chosen to control the autonomous vehicles. Compared with other methods, such as PID, MPC, etc., what are the main advantages of fuzzy control?
(2) Compared with type-1 fuzzy control, why type-2 fuzzy control is more suitable for autonomous vehicles control?
(3) What are the main novelty of the study?
(4) In the experiment sections, more comparative studies should be conducted with the presented method, e.g., PID, MPC, etc.
(5) More evaluation metrics should be calculated to show the performance of the comparison methods, e.g., yaw rate, comfortability, etc.
(6) The language should be polished carefully. And a lot of typos should be fixed.
Reviewer 3 Report
1. It is suggested that the authors should include the following study in the literature https://www.sciencedirect.com/science/article/abs/pii/S0045790617336340
2. The author should explain Figure 2, Figure 6, Figure 9 and Figure 10 more illustratively.
3. It is suggested that the authors should mention about computational platform where the FLC has been designed along with the computational time elapsed in the process.
4. What is the limitation of the proposed study? Discuss it in a separate paragraph.
5. The authors should list the various notations used in their study in an Appendix section.
6. The future scope of the study should be mentioned in a separate paragraph under the Conclusion section.
Round 2
Reviewer 1 Report
A state-space state equation is of the form dX/dt=AX+Bu. The authors should refer to any textbook of automatic control systems. The state-space equation on line 634 is incorrect. The left-hand side of the equation is a vector while the right-hand side is a matrix plus a vector. The right-hand side of the equation apparently cannot be written as the right-hand side of the equation. Since no indications give \theta_vf in (24)=\theta_yf in (18) and \theta_vr in (25)=\theta_yf in (19), we cannot substitute eqns. (18),(19), (24), (25) into eqns. (16), (17). Eqn. (21) seems incorrect. It should be (\theta_vr rather than \theta_vf). In summary, the authors should re-check eqns. (16)-(25) (including notation consistency), derive and write a correct state-space state equation.
Reviewer 2 Report
This manuscript has been improved based on the last round reviewed results basically. The presentation of the study should be improved more carefully, e.g., the Fig. 9 and 10 are very not clear, there are still some typos in the manuscript.
Reviewer 3 Report
The article has been revised and improved based on previous comments. This paper can be accepted for publication after some minor revision.
- The Introduction section could have been written in a more structured way. Related work may be given separately.
- The problem statement may be elaborated further with recent literature review and more clarification. Authors can include the following:
(i) https://onlinelibrary.wiley.com/doi/full/10.1002/2050-7038.12875
(ii) https://www.scirp.org/html/3-1680302_106367.htm
